# Pneumococcal meningitis and endotoxemia: A cross-sectional clinical study

**Sylvie Nguyen[1], Jeanne Godon[2], Jean-Paul Pais de Barros[3], David Masson[1], Lionel Piroth[2,4], Mathieu Blot** [1,2,4*], **Meningotoxemia collaboration group[2¶]**

**1** Lipness Team, INSERM Research Centre LNC-UMR1231 and LabEx LipSTIC, University of Burgundy, Dijon, France, **2** Department of Infectious Diseases, Dijon-Bourgogne University Hospital, Dijon, France, **3** Lipidomic Analytic Platform, Dijon, France, **4** CHU Dijon-Bourgogne, INSERM, Université de Bourgogne, CIC, Module Épidémiologie Clinique, Dijon, France and LabEx LipSTIC, University of Burgundy, Dijon, France

¶ Co-authors of the Meningotoxemia collaboration group are listed in the Acknowledgements section
* mathieu.blot@chu-dijon.fr

## Abstract

### Introduction

Lipopolysaccharide (LPS) is a major virulence factor during both meningococcal and *Haemophilus influenzae* meningitis. Pneumococcus does not produce LPS but could be responsible for bacterial digestive translocation as a consequence of sepsis. We addressed this question in the context of pneumococcal meningitis.

### Methods

A cross-sectional study on 24 patients with pneumococcal meningitis (20 (83%) admitted in intensive care unit, 4 (17%) with septic shock) and 34 prospectively-enrolled healthy volunteers. Interleukin 6 and C-reactive proteins plasma concentrations were measured as markers of systemic inflammation. Endotoxemia was measured using mass spectrometry (LC-MS/MS) for detection of molecules bound to the lipid A, namely 3-OH fatty acids.

### Results

Meningitis patients had significantly higher levels of plasma C-reactive protein (237 (74–373) vs. 2 (2–2) mg/l, $p < 0.001$ and interleukin 6 (43 (13–128) vs. 4.6 (4.6–16.6) pg/ml; $p < 0.001$) than healthy volunteers. However, we observed no significant difference in plasma lipopolysaccharide concentrations between patients and healthy volunteers (674 (554–896) vs. 668 (623–777) pmol/ml; $p = 0.546$).

### Conclusions

Our results suggest that LPS is not a key determinant of the excessive inflammation associated with severe forms of pneumococcal meningitis.

**Data availability statement:** The raw data concerning LPS and immune biomarkers concentrations are available in the supplemental file.

**Funding:** This work was supported by a grant from the French National Hospital Clinical Research Program [PHRC] 2004/37 (HyaloStrepto project), a grant from AOI (Appel d'Offre Interne) from the University hospital of Dijon, and from the INSERM (Institut National de la Sante et de la Recherche Médicale - Center de Recherche UMR 1231, Dijon, France), the national research agency (ANR) Investissements d'Avenir Grant (ANR-11 LABX-0021-01, Labex Lipstic, Dijon, France), and the Université Bourgogne Franche Comte (Dijon, France). The funders had no role in study design, data collection and analysis, decision to publish, or preparation of the manuscript.

**Competing interests:** All the authors have no conflicts of interest to report.

**Abbreviations:** CRP, C-reactive protein; *H. influenzae*, *Haemophilus influenzae*; i-FABP, intestinal-Fatty-acid-binding protein; IL-6, interleukin 6; LC-MS/MS, liquid chromatography coupled to tandem mass spectrometry; LPS, lipopolysaccharide; *N. meningitidis*, *Neisseria meningitidis*; *S. pneumoniae*, *Streptococcus pneumoniae*.

## Introduction

Bacterial meningitis is a devastating disease with a high case morbidity and fatality worldwide despite appropriate antibiotic therapies. *Streptococcus pneumoniae* (*S. pneumoniae*), *Neisseria meningitidis* (*N. meningitidis*) and *Haemophilus influenzae* (*H. influenzae*) are the three leading causes of bacterial meningitis in adults. Inflammatory response plays a key role in pathogenesis and is initiated through the recognition of virulence factors. Lipopolysaccharide (LPS) is one of the major virulence factors of *N. meningitis* and *H. influenzae*, and plays a key role in the induction of septic shock [1]. While *S. pneumoniae* does not express LPS, evidence suggests that sepsis promotes gut barrier dysfunction, which releases large amounts of LPS into host blood [2,3]. In a vicious cycle, this may in turn influence a systemic inflammatory cascade, leading to multiple organ dysfunction and/or death [4,5]. There are biomarkers capable of reflecting this intestinal dysfunction. For example, intestinal fatty acid binding protein (I-FABP) is a marker of intestinal damage (ID), released into circulation when the integrity of enterocyte membranes in the small intestine is compromised. Primarily expressed in enterocytes, I-FABP plays a critical role in fatty acid transport. Elevated plasma I-FABP levels indicate intestinal mucosal injury or ischemia and are used as a biomarker in conditions like inflammatory bowel disease and acute intestinal ischemia. In healthy individuals, plasma I-FABP levels typically range between 200–600 pg/ml [6,7].

As patients suffering from bacterial meningitis often present with sepsis, our study aimed to investigate if pneumococcal meningitis is associated with high circulating levels of LPS in a cohort of patients, compared with healthy volunteers. If the hypothesis is confirmed, it could offer new therapeutic avenues, such as LPS apheresis or the administration of recombinant plasma phospholipid transfer protein, which is likely to remove LPS or to inactivate it.

## Materials and methods

We conducted a cross-sectional study including 24 patients with pneumococcal meningitis from a longitudinal, observational, prospective cohort study (Hyalo-Strepto project, NCT01931800) approved by the National Commission on Informatics and Liberty and by the local ethics committee (Comité de Protection des Personnes Est I). Oral consent was obtained from the patient or their legal representative. From January 1, 2005 to December 31, 2022, at the Dijon-Bourgogne University hospital, patients meeting the following 3 criteria were included: (i) aged 18 years or older who provided oral consent; (ii) diagnosis of acute meningitis defined as an acute illness with typical clinical signs (headaches, neck stiffness, sensitivity to light, nausea, confusion) and CSF pleiocytosis (≥5/mm3), and (iii) positive CSF culture and/or blood culture for *S. pneumoniae*. Whole blood was collected at admission using citrate tube, once the result of positive culture was communicated, and the plasma was stored.

We included 34 healthy volunteers (HV) between April and June 2022, from an observational, prospective cohort study (Lymphonie project, NCT03505281) that was approved by the local ethics committee (Comité de Protection des Personnes SUD

MEDITERRANEE V; 2017-A03404-49). Inclusion criteria were: (i) individuals aged 18 years or older who provided oral consent; (ii) absence of fever, antibiotic use or surgery in the last 30 days, (iii) temperature < 37.8°C the day of inclusion, absence of suspected infection, (iv) no immune deficiency. Citrated whole blood was obtained and the plasma was stored.

Data were collected using standardized report forms and were fully anonymized before analyses. Comorbid conditions, including those that could lead to intestinal ischemia, were rigorously assessed.

Human plasma IL-6 and intestinal-fatty-acid-binding protein (i-FABP; considered as a biomarker of intestinal damage) concentrations were determined with an ELISA assay (R&D Systems, Minneapolis, MN, USA).

LPS plasma concentrations were determined using a patented high performance mass spectrometry method (liquid chromatography coupled with tandem mass spectrometry (LC-MS/MS)) (EndoQuant) for detection of 3-OH fatty acids (C10, C12, C14, C16 and C18), which are molecules bound to the lipid A motif of LPS [8,9]. Briefly, plasma was spiked with an internal standard (3-hydroxydecanoic, 3-hydroxydodecanoic, 3-hydroxytridecanoic, 3-hydroxyhexadecanoic and 3-hydroxyoctadecdienoic acids). Samples were hydrolysed in 8-M hydrochloric acid for 3 h at 90 °C. Free fatty acids were extracted with distilled water and a mix of hexane/ ethyl acetate (3/2 v/v). The organic phase was discarded by evaporation. The dried extracts were solubilized in ethanol and injected on a ZORBAX SB-C18 column (Agilent) coupled with an infinity 1260 HPLC binary system (Agilent) for fatty acid separation. MS/MS detection was performed using a QQQ 6490 triple quadruple mass spectrometer (Agilent). 3OH-fatty acids quantitation was performed by negative SRM mode as previously described [8,10]. The esterified form of 3-OH fatty acids (bound to LPS) was calculated as the total-free concentrations. Total LPS concentrations were the sum of each esterified form of 3-OH fatty acids concentrations.

Qualitative variables were presented as numbers (percentage) and compared using the $\chi 2$ test (or Fisher's exact test when appropriate). Continuous variables were presented as medians (interquartile range, IQR) and compared with the Mann-Whitney U-test. A $p$ value lower than 0.05 was considered statistically significant. All statistical analyses were performed using JASP (an open-source statistics program (University of Amsterdam, The Netherlands) [11] or Prism software (GraphPad Prism)).

## Results

The current analysis included 24 patients with culture-proven pneumococcal meningitis (positive CSF for 23/24, and decapitated meningitis with positive blood culture for one) and 34 HV, none of whom had conditions predisposing to intestinal ischemia.

Characteristics of patients and HV are reported in Table 1. Median (interquartile range) age was 62 (49–70) years for patients and 63 (52–71) for HV with no significant difference (p = 0.693), and sex ratio was not significantly different between the two groups (p = 0.724). Twenty (83%) patients were admitted to the ICU, four (17%) had septic shock and 30-day mortality was 25%.

Meningitis patients had significantly higher levels of blood leukocytes (15.9 (14.3–20.2) vs. 6.3 ((5.2–7.3); p < 0.001), plasma C-reactive protein (237 (74–373) vs. 2 (2–2) mg/l, p < 0.001) and plasma interleukin 6 (43 (13–128) vs. 4.6 (4.6–16.6) pg/ml; p < 0.001) than healthy volunteers (Table 1, Fig 1A, Fig 1B).

In contrast, plasma LPS concentrations were not significantly different between patients and volunteers (674 (554–896) vs. 668 (623–777) pmol/ml; p = 0.546) (Fig 1C). Plasma concentrations of iFABP were, however, significantly lower in patients compared with volunteers (Fig 1D).

## Discussion

Here we demonstrate that pneumococcal meningitis is not linked to elevated circulating endotoxin levels compared to healthy volunteers. The gastrointestinal tract contains trillions of microorganisms, mainly Gram-negative bacteria, that exist symbiotically with a tolerant intestinal immune system. In sepsis, the intestinal dysfunction that occurs could be associated with endotoxemia through bacterial translocation, enhancing inflammatory response [5,10]. A preclinical study showed that

**Table 1. Characteristics of healthy volunteers and patients with pneumococcal meningitis (Meningotoxemia study, 2023).**

| | Missing data (HV/PM) | Healthy volunteers n = 34 | Pneumococcal meningitis n = 24 | p-value |
|---|---|---|---|---|
| **Demographic data** | | | | |
| Age, median (IQR) | 0 | 63 (52-71) | 62 (49-70) | 0.693 |
| Sex (Male), n (%) | 0 | 20 (59) | 13 (54) | 0.724 |
| **Comorbid conditions** | | | | |
| Alcohol active use, n (%) | 0 | 0 | 3 (13) | 0.130 |
| Immunodepression, n (%) | 0 | 0 | 7 (29) | 0.003 |
| Chronic pulmonary disease, n (%) | 0 | 0 | 2 (8) | 0.326 |
| Chronic heart disease, n (%) | 0 | 1 (3) | 6 (25) | 0.033 |
| Chronic renal insufficiency | 0 | 0 | 2 (8) | 0.326 |
| Diabetes mellitus, n (%) | 0 | 1 (3) | 3 (13) | 0.374 |
| Cirrhosis, n (%) | 0 | 0 | 1 (4) | 0.860 |
| **Clinical features** | | | | |
| Temperature (°C), median (IQR) | 1/1 | 36.0 (35.8-36.3) | 38.6 (38.2-39.4) | < 0.001 |
| Glasgow score, median (IQR) | 0 | 15 (15-15) | 9 (7-11) | < 0.001 |
| Systolic arterial pressure < 100 mm Hg, n (%) | 0 | 1 (3) | 6 (25) | 0.033 |
| Respiratory rate ≥ 30/min, n (%) | 0/3 | 0 | 6 (29) | 0.004 |
| Septic shock, n (%) | 0 | 0 | 4 (17) | 0.052 |
| **Biological features** | | | | |
| Hemoglobin (mg/dl), median (IQR) | 0/8 | 14.7 (13.3-15.5) | 13.1 (13.3-15.6) | 0.004 |
| Leukocytes (x$10^6$/l), median (IQR) | 0/6 | 6.3 (5.2-7.3) | 15.9 (14.3-20.2) | < 0.001 |
| Platelets (/mm3), median (IQR) | 0/7 | 232 (198-268) | 214 (151-261) | 0.181 |
| Neutrophils (x$10^6$/l), median (IQR) | 0/9 | 3.6 (2.9-4.3) | 14.4 (12.0-17.3) | < 0.001 |
| Lymphocytes (x$10^3$/l), median (IQR) | 0/10 | 1625 (1328-2148) | 680 (515-780) | < 0.001 |
| Monocytes (x$10^3$/l), median (IQR) | 0/10 | 420 (348-593) | 900 (747-1193) | < 0.001 |
| C-reactive protein (mg/l), median (IQR) | 0/5 | 2 (2-2) | 237 (92-357) | < 0.001 |
| **Microbiological features** | | | | |
| Positive blood cultures for S. pneumoniae, n (%) | NA/4 | NA | 17 (85) | |
| Positive CSF cultures for S. pneumoniae, n (%) | 0 | NA | 23 (96) | |
| CSF white blood cell count (cells/µl) | NA/4 | NA | 1140 (588-3125) | |
| CSF % of neutrophils (%) | NA/6 | NA | 92 (81-95) | |
| CSF protein concentration (g/L) | NA/6 | NA | 6.7 (3.1-7.8) | |
| **Outcomes** | | | | |
| Intensive Care Unit admission, n (%) | 0 | NA | 20 (83) | |
| Mechanical ventilation, n (%) | 0 | NA | 17 (71) | |
| 30-day mortality, n (%) | 0 | NA | 6 (25) | |
| 90-day mortality, n (%) | 0 | NA | 6 (25) | |
| **Plasma biological investigations** | | | | |
| LPS (pg/ml), median (IQR) | 0 | 668 (623-777) | 674 (554-896) | 0.546 |
| IL-6 (pg/ml), median (IQR) | 0 | 4.6 (4.6-16.6) | 43 (13-128) | < 0.001 |
| iFABP (pg/ml), median (IQR) | 0 | 432 (248-651) | 215 (99-300) | < 0.001 |

Abbreviations: CSF: cerebro-spinal fluid, iFABP: intestinal Fatty Acid Binding Protein, IL: interleukin, IQR: interquartile range, LPS: lipopolysacharide, NA: not applicable

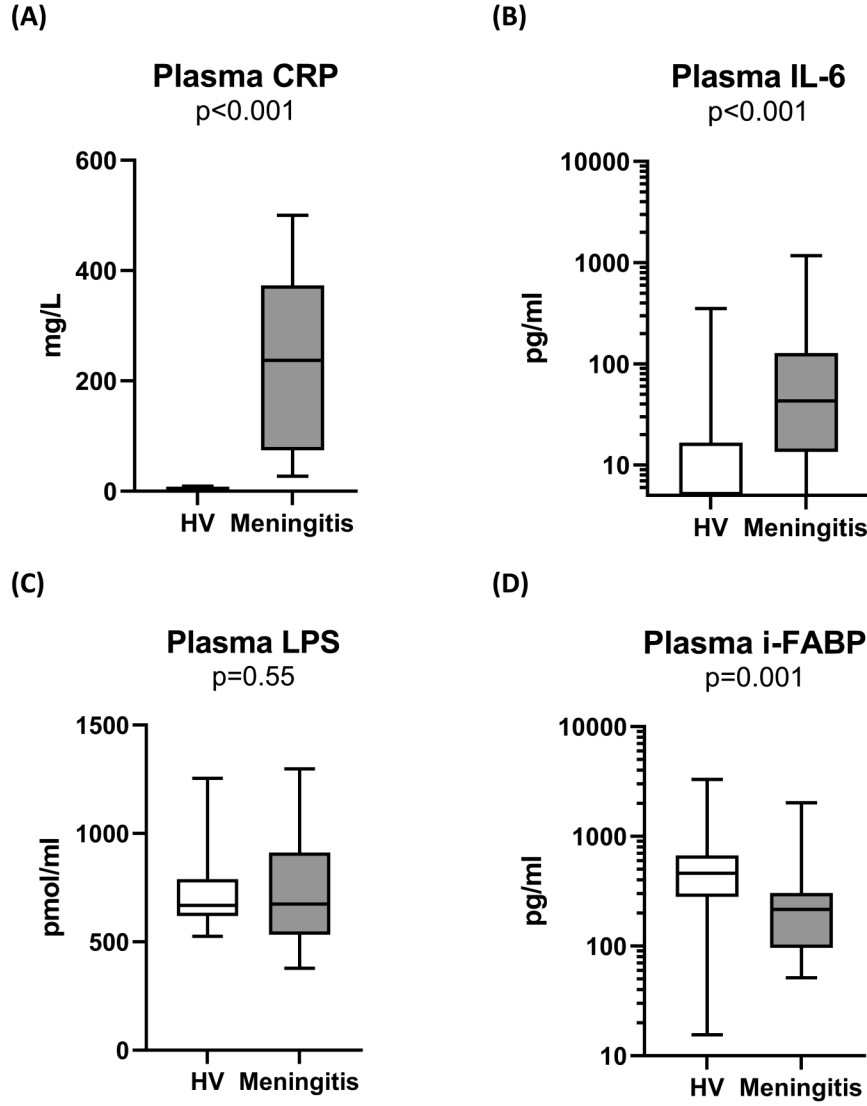

**Fig 1. Plasma inflammatory response, LPS concentrations, and marker of gut barrier integrity in 24 patients prospectively included with a pneumococcal meningitis and 34 healthy volunteers (Meningotoxemia study, 2023).** Plasma C-reactive protein **(A)**, IL-6 **(B)**, LPS **(C)** and i-FABP **(D)** concentrations were measured. LPS concentrations were measured using a mass spectrometry (LC-MS/MS) patented method for detection of 3-OH fatty acids (C10, C12, C14, C16 and C18), which are molecules bound to the lipid A motif of LPS.Abbreviations: CRP: C-reactive protein, HV: healthy volunteers, i-FABP: intestinal-Fatty-acid-binding protein, IL-6: interleukin 6, LC-MS/MS: liquid chromatography coupled to tandem mass spectrometry, LPS: lipopolysaccharide.

pneumococcal sepsis was associated with translocation of peptidoglycan from the gut into systemic circulation and likely to prime neutrophil function [3]. Pneumococcal meningitis is a model of choice to test this hypothesis since *S. pneumoniae* does not produce LPS. In addition, pneumococcal meningitis is associated with high mortality, as confirmed in our study with a 25% 30-day mortality, and the majority of patients being admitted to ICU with multiple organ dysfunction. Finally, it is also associated with an intense systemic inflammatory response, as confirmed in our study by the high level of C-reactive protein, IL-6 and neutrophils, which are well known biomarkers of LPS-induced inflammation. However, our data does not support the "intestinal hypothesis" since the plasma LPS concentrations were not higher in pneumococcal meningitis, and

neither was plasma iFABP. In addition, we observed that pneumococcal pneumonia was not associated with enhanced plasma LPS concentrations in a preclinical rabbit model or when patients were compared to HV [7].

As previously discussed, quantifying LPS is challenging, leading to discrepancies depending on the method used, particularly with regard to false positives in the LAL assay [9]. To address this, we employed a validated method to measure the most abundant 3-OH-fatty acids in lipid A, enabling sensitive, accurate, and direct quantification of total LPS levels in plasma [8,10,12].

In addition, we observed that the surrogate marker of gut leakage, namely iFABP concentrations, were actually lower in meningitis patients than in healthy volunteers. This has also been observed in the context of pneumonia, whether of viral (COVID-19) or bacterial (*S. pneumoniae*) origin [9,13]. Other studies have also demonstrated higher concentrations of iFABP in healthy volunteers compared to patients with mesenteric ischemia, a condition well-known for its association with intestinal injury [14]. Additionally, no underlying diseases causing intestinal ischemia were identified in healthy individuals. However, these discrepancies between studies could be attributed to the different tools used to measure iFABP, or the timing of the iFABP measurement. In our study, sample from meningitis patients were collected as soon as the pneumococcal strain was identified, approximately two days after admission. We cannot rule out the possibility of an earlier increase in iFABP levels occurring immediately upon admission, followed by a subsequent decrease, as iFABP concentrations may decline with the resolution of intestinal damage.

If our study does not support the intestinal hypothesis, high levels of systemic inflammation are primarily related to the host's response against the pathogen and its numerous virulence factors, including pneumolysin and the capsule, during invasive pneumococcal infection [1,4].

Our study has several limits. First, we included a small number of patients, but pneumococcal meningitis is infrequent and this type of prospective cohort with biobanking is rare. Then, we observed a difference between patients and volunteers due to the immunocompromised status of some patients, which could bias the interpretation of plasma endotoxin concentrations. However, as immunocompromised patients are more susceptible to bacterial translocation, this bias can be considered conservative, and not an issue to interpret our data as LPS concentrations were not increased in patients compared to healthy volunteers. Finally, our method only measured LPS mass, but we did not exclude an increased LPS activity in plasma.

In conclusion, endotoxin levels were unchanged in pneumococcal meningitis. Unlike *N. meningitidis* or *H. influenzae* meningitis, these results indicate that LPS does not drive the excessive systemic inflammatory response and the resulted multiorgan dysfunction in pneumococcal meningitis.

## Supporting information

**S1 Data. Supplemental data raw data of the 34 healthy volunteers (HV) and the 24 patients concerning plasma LPS, C-reactive protein (CRP), interleukin 6 (IL-6) and intestinal-Fatty-acid-binding protein (iFABP) plasma concentrations.**
(XLSX)

## Acknowledgments

The authors thank Julie Quantin from the CIC-P for the inclusion of healthy volunteers, Sandrine Gohier, Carole Charles and Delphine Croisier from the Infectious department for the inclusion of patients. We thank Maud Carpentier from the DRCI (Direction de la Recherche Clinique et de l'Innovation), and Suzanne Rankin for proofreading and editing the manuscript.

**Meningotoxemia collaboration group (lead author: Mathieu Blot;** mathieublot2@gmail.com**):**

Delphine Croisier (Vivexia, Dijon, France), Sandrine Gohier (Infectious Department, Dijon University Hospital, Dijon, France), Carole Charles (Infectious Department, Dijon University Hospital, Dijon, France), Marc Bardou (INSERM CIC-P 1432, Dijon University Hospital, Dijon, France), Ines Ben Guezala (INSERM CIC-P 1432, Dijon University Hospital, Dijon,

France), Jeanne Godon (Infectious Department, Dijon University Hospital, Dijon, France), Pierre-Emmanuel Charles (Intensive Care Unit, Dijon University Hospital, Dijon, France), Sylvie Nguyen (Lipness Team, INSERM Research Centre LNC-UMR1231), Thomas Gautier (Lipness Team, INSERM Research Centre LNC-UMR1231), Jean-Paul Pais de Barros (Lipidomic Analytic Platform, Dijon, France), Hélène Choubley (Lipidomic Analytic Platform, Dijon, France), Victoria Bergas (Lipidomic Analytic Platform, Dijon, France), Marine Jacquier (Intensive Care Unit, Dijon University Hospital, Dijon, France), Jennifer Tetu (Laboratory of bacteriology, Dijon University Hospital, Dijon, France), Jean-Pierre Quenot (Intensive Care Unit, Dijon University Hospital, Dijon, France), Maxime Luu (INSERM CIC-P 1432, Dijon University Hospital, Dijon, France), Emilie Galizzi (INSERM, CIC 1432, Clinical Epidemiology unit, Dijon, France), Christine Binquet (INSERM, CIC 1432, Clinical Epidemiology unit, Dijon, France), David Masson (Lipness Team, INSERM Research Centre LNC-UMR1231), Lionel Piroth (Infectious Department, Dijon University Hospital, Dijon, France), Mathieu Blot (Infectious Department, Dijon University Hospital, Dijon, France).

## Author contributions

**Conceptualization:** Sylvie Nguyen, Jeanne Godon, Mathieu Blot.

**Data curation:** Mathieu Blot.

**Formal analysis:** Sylvie Nguyen, Jeanne Godon.

**Funding acquisition:** David Masson, Lionel Piroth, Mathieu Blot.

**Investigation:** Sylvie Nguyen, Jeanne Godon, Jean-Paul Pais de Barros, Meningotoxemia collaboration group ., Mathieu Blot.

**Methodology:** Lionel Piroth, Mathieu Blot.

**Supervision:** Lionel Piroth, Mathieu Blot.

**Writing – original draft:** Sylvie Nguyen, Jeanne Godon, Mathieu Blot.

**Writing – review & editing:** Jean-Paul Pais de Barros, David Masson, Lionel Piroth.

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
