## [Decision Letter · Decision Letter 0]

12 Dec 2024

PONE-D-24-44702Pneumococcal meningitis and endotoxemia: a cross-sectional clinical studyPLOS ONE

Dear Dr. Blot,

Thank you for submitting your manuscript to PLOS ONE. After careful consideration, we feel that it has merit but does not fully meet PLOS ONE’s publication criteria as it currently stands. Therefore, we invite you to submit a revised version of the manuscript that addresses the points raised during the review process.

We look forward to receiving your revised manuscript.

Kind regards,

Muhammad Afzal, PhD

Academic Editor

PLOS ONE

Journal Requirements:

When submitting your revision, we need you to address these additional requirements. 1. Please ensure that your manuscript meets PLOS ONE's style requirements, including those for file naming. The PLOS ONE style templates can be found at https://journals.plos.org/plosone/s/file?id=wjVg/PLOSOne_formatting_sample_main_body.pdf and https://journals.plos.org/plosone/s/file?id=ba62/PLOSOne_formatting_sample_title_authors_affiliations.pdf 2. Your abstract cannot contain citations. Please only include citations in the body text of the manuscript, and ensure that they remain in ascending numerical order on first mention. 3. Please match your authorship list in your manuscript file and in the system. 4. PLOS requires an ORCID iD for the corresponding author in Editorial Manager on papers submitted after December 6th, 2016. Please ensure that you have an ORCID iD and that it is validated in Editorial Manager. To do this, go to ‘Update my Information’ (in the upper left-hand corner of the main menu), and click on the Fetch/Validate link next to the ORCID field. This will take you to the ORCID site and allow you to create a new iD or authenticate a pre-existing iD in Editorial Manager. 5. We noticed you have some minor occurrence of overlapping text with the following previous publication(s), which needs to be addressed: - https://doi.org/10.1111/eci.14077 In your revision ensure you cite all your sources (including your own works), and quote or rephrase any duplicated text outside the methods section. Further consideration is dependent on these concerns being addressed. 6. We note that the grant information you provided in the ‘Funding Information’ and ‘Financial Disclosure’ sections do not match.  When you resubmit, please ensure that you provide the correct grant numbers for the awards you received for your study in the ‘Funding Information’ section. 7. Thank you for stating the following in your Competing Interests section:  [none to disclose]. Please complete your Competing Interests on the online submission form to state any Competing Interests. If you have no competing interests, please state ""The authors have declared that no competing interests exist."", as detailed online in our guide for authors at http://journals.plos.org/plosone/s/submit-now  This information should be included in your cover letter; we will change the online submission form on your behalf. 8. Please provide a complete Data Availability Statement in the submission form, ensuring you include all necessary access information or a reason for why you are unable to make your data freely accessible. If your research concerns only data provided within your submission, please write "All data are in the manuscript and/or supporting information files" as your Data Availability Statement. 9. One of the noted authors is a group [Meningotoxemia collaboration group]. In addition to naming the author group, please list the individual authors and affiliations within this group in the acknowledgments section of your manuscript. Please also indicate clearly a lead author for this group along with a contact email address.

Reviewers' comments:

Reviewer's Responses to Questions

**Comments to the Author**

1. Is the manuscript technically sound, and do the data support the conclusions?

Reviewer #1: Partly

Reviewer #2: Yes

2. Has the statistical analysis been performed appropriately and rigorously? 

Reviewer #1: I Don't Know

Reviewer #2: Yes

3. Have the authors made all data underlying the findings in their manuscript fully available?

Reviewer #1: Yes

Reviewer #2: Yes

4. Is the manuscript presented in an intelligible fashion and written in standard English?

Reviewer #1: Yes

Reviewer #2: Yes

5. Review Comments to the Author

Reviewer #1: Dear Authors,

Where is intestinal Fatty Acid Binding Protein found, what functions does it perform, what does its height indicate. Which diseases are valuable in measuring intestinal Fatty Acid Binding Protein? Is there a defined normal value for intestinal Fatty Acid Binding Protein? It is recommended to specify in the introduction section.

How do you explain the finding of higher intestinal Fatty Acid Binding Protein levels in healthy individuals, contrary to the hypothesis of your study?

Are you sure that there is no disease that causes intestinal ischemia in individuals defined as healthy individuals? If you are sure, please state this in the material method

Reviewer #2: General comment – This is a very interesting research question that aims to define a role, if any, of LPS in the systemic inflammatory response during pneumococcal meningitis. The study is well written and diligently conducted. The results are significant and relevant, adding good information for understanding the pathophysiology of this disease.

There are some points the authors should review, as follows

1. In Introduction section correct, in Line 3, Neisseria meningitidis (N. meningitidis)

2. Which was the time frame for including healthy volunteers? Describe it in the manuscript

3. The dried extract extracts - remove the duplicate word

4. “Then, we observed a difference between patients and volunteers, considering immunocompromised status, which may introduce a potential bias in the interpretation of plasma endotoxin concentrations. Since immunocompromised patients are susceptible to gut translocation, this can nevertheless be considered as a conservative bias”- I stopped at this sentence. I read it and reread it several times, but I didn't really understand what you meant by it. Would you please rephrase it? The difference observed for biomarkers was expected.

6. PLOS authors have the option to publish the peer review history of their article (what does this mean? ). If published, this will include your full peer review and any attached files.

**Do you want your identity to be public for this peer review?** For information about this choice, including consent withdrawal, please see our Privacy Policy .

Reviewer #1: No

Reviewer #2: No

---

## [Author Response · Author response to Decision Letter 0]

14 Mar 2025

Dear Editor,

Please find enclosed our revised article entitled “Pneumococcal meningitis and endotoxemia: a cross-sectional clinical study”, by Sylvie Nguyen et al.

We would like to thank the reviewers for all the comments and advice that we carefully followed in order to improve the manuscript. Changes and revisions requested are highlighted in this new version of our manuscript. A point-by-point reply to the reviewer that indicates how the manuscript has been revised is provided below. We have also addressed all the journal requirements.

• Why is this manuscript suitable for publication in PLoS ONE?

This is a prospective study with an original investigation: measuring endotoxemia in the context of pneumococcal meningitis.

• How does your paper provide a worthwhile addition to the scientific literature?

Our results suggest that LPS is not a key determinant of the excessive inflammation associated with severe forms of pneumococcal meningitis.

• How does your paper relate to previously published work?

Previous data suggest that gut translocation can lead to endotoxemia during sepsis, even when the origin of the sepsis is not related to a Gram-negative bacterium, a known source of LPS. We tested this hypothesis in the context of pneumococcal meningitis.

• Which types of scientists do you believe will be most interested in your study?

This study is likely to interest clinicians specializing in infectious diseases and critical care, as well as microbiologists and scientists focused on endotoxemia.

We hope that you will find this new version of our article suitable for publication in your journal.

Yours sincerely,

Mathieu Blot

o Response: We ensure that our manuscript meets PLOSE ONE’s style requirements

2. Your abstract cannot contain citations. Please only include citations in the body text of the manuscript, and ensure that they remain in ascending numerical order on first mention.

o Response: Citations have been only cited in the body of text of the manuscript

3. Please match your authorship list in your manuscript file and in the system.

o Response: The meningotoxemia collaboration group should be added in the authorship list. The list of authors is listed a the end of the manuscript with additional information requested:

o Response: The ORCID number has been indicated

5. We noticed you have some minor occurrence of overlapping text with the following previous publication(s), which needs to be addressed:

- https://doi.org/10.1111/eci.14077

In your revision ensure you cite all your sources (including your own works), and quote or rephrase any duplicated text outside the methods section. Further consideration is dependent on these concerns being addressed.

o Response: We have reviewed the overlapping text with the referenced publication (https://doi.org/10.1111/eci.14077) and have made the necessary revisions. We have ensured that all sources, including our own works, are properly cited, and any duplicated text outside the methods section has been either quoted or rephrased. We believe these changes address your concerns and look forward to your further consideration.

o Response: We added the funding received by the “Université Bourgogne Franche Comte”. This is not a grant specifically received for this project, but ongoing financial support. Thus, we do not have award number.

Funding section:

This work was supported by a grant from the French National Hospital Clinical Research Program [PHRC] 2004/37 (HyaloStrepto project), a grant from AOI (Appel d’Offre Interne) from the University hospital of Dijon, and from the INSERM (Institut National de la Sante et de la Recherche Médicale - Center de Recherche UMR 1231, Dijon, France), the national research agency (ANR) Investissements d’Avenir Grant (ANR-11 LABX-0021-01, Labex Lipstic, Dijon, France), and the Université Bourgogne Franche Comte (Dijon, France).

7. Thank you for stating the following in your Competing Interests section:

[none to disclose].

Response: I cannot find the section online corresponding to 'competing interests which are: All the authors have no conflicts of interest to report.

8. Please provide a complete Data Availability Statement in the submission form, ensuring you include all necessary access information or a reason for why you are unable to make your data freely accessible. If your research concerns only data provided within your submission, please write "All data are in the manuscript and/or supporting information files" as your Data Availability Statement.

o Response: we modified the statement : “The data underlying the results presented in the study are available from mathieu.blot@chu-dijon.fr”

9. One of the noted authors is a group [Meningotoxemia collaboration group]. In addition to naming the author group, please list the individual authors and affiliations within this group in the acknowledgments section of your manuscript. Please also indicate clearly a lead author for this group along with a contact email address.

o Response: We modified the group as indicated :

Meningotoxemia collaboration group (lead author: Mathieu Blot; mathieublot2@gmail.com):

Delphine Croisier (Vivexia, Dijon, France), Sandrine Gohier (Infectious Department, Dijon University Hospital, Dijon, France), Carole Charles (Infectious Department, Dijon University Hospital, Dijon, France), Marc Bardou (INSERM CIC-P 1432, Dijon University Hospital, Dijon, France), Ines Ben Guezala (INSERM CIC-P 1432, Dijon University Hospital, Dijon, France), Jeanne Godon (Infectious Department, Dijon University Hospital, Dijon, France), Pierre-Emmanuel Charles (Intensive Care Unit, Dijon University Hospital, Dijon, France), Sylvie Nguyen (Lipness Team, INSERM Research Centre LNC-UMR1231), Thomas Gautier (Lipness Team, INSERM Research Centre LNC-UMR1231), Jean-Paul Pais de Barros (Lipidomic Analytic Platform, Dijon, France), Hélène Choubley (Lipidomic Analytic Platform, Dijon, France), Victoria Bergas (Lipidomic Analytic Platform, Dijon, France), Marine Jacquier (Intensive Care Unit, Dijon University Hospital, Dijon, France), Jennifer Tetu (Laboratory of bacteriology, Dijon University Hospital, Dijon, France), Jean-Pierre Quenot (Intensive Care Unit, Dijon University Hospital, Dijon, France), Maxime Luu (INSERM CIC-P 1432, Dijon University Hospital, Dijon, France), Emilie Galizzi (INSERM, CIC 1432, Clinical Epidemiology unit, Dijon, France), Christine Binquet (INSERM, CIC 1432, Clinical Epidemiology unit, Dijon, France), David Masson (Lipness Team, INSERM Research Centre LNC-UMR1231), Lionel Piroth (Infectious Department, Dijon University Hospital, Dijon, France), Mathieu Blot (Infectious Department, Dijon University Hospital, Dijon, France).

Thank you for stating the following financial disclosure:

"This work was supported by a grant from the French National Hospital Clinical Research Program [PHRC] 2004/37 (HyaloStrepto project), a grant from AOI (Appel d’Offre Interne) from the University hospital of Dijon, and from the INSERM (Institut National de la Sante et de la Recherche Médicale - Center de Recherche UMR 1231, Dijon, France), the national research agency (ANR) Investissements d’Avenir Grant (ANR-11 LABX-0021-01, Labex Lipstic, Dijon, France), and the Université Bourgogne Franche Comte (Dijon, France)."

o Response: we added this statement in our manuscript: "The funders had no role in study design, data collection and analysis, decision to publish, or preparation of the manuscript."

3. In line with our goal of ensuring long-term data availability to all interested researchers, PLOS’ Data Policy states that authors cannot be the sole named individuals responsible for ensuring data access (http://journals.plos.org/plosone/s/data-availabilityloc-acceptable-data-sharing-methods). In light of this, please consider whether any of the following would be possible:

Can you make a de-identified, anonymized, or aggregated data set publicly available in a repository or, if that is not possible, in your manuscript’s Supporting Information files? For a list of recommended repositories, some of which are able to hold sensitive data, see here:

o Response: As requested, raw data concerning LPS and immune biomarkers concentrations has been added in a supplemental file.

Response to Reviewer #1:

We would like to thank the reviewer 1 for all the comments and advice that we carefully followed in order to improve the manuscript

1. "Where is intestinal Fatty Acid Binding Protein found, what functions does it perform, what does its height indicate. Which diseases are valuable in measuring intestinal Fatty Acid Binding Protein? Is there a defined normal value for intestinal Fatty Acid Binding Protein? It is recommended to specify in the introduction section."

o Response: We have added detailed information in the Introduction section regarding the location, functions, and clinical relevance of iFABP.

" There are biomarkers capable of reflecting this intestinal dysfunction. For example, intestinal fatty acid binding protein (I-FABP) is a marker of intestinal damage (ID), released into circulation when the integrity of enterocyte membranes in the small intestine is compromised. Primarily expressed in enterocytes, I-FABP plays a critical role in fatty acid transport. Elevated plasma I-FABP levels indicate intestinal mucosal injury or ischemia and are used as a biomarker in conditions like inflammatory bowel disease and acute intestinal ischemia. In healthy individuals, plasma I-FABP levels typically range between 200-600 pg/ml (1,2)"

2. "How do you explain the finding of higher intestinal Fatty Acid Binding Protein levels in healthy individuals, contrary to the hypothesis of your study?"

o Response: In the Discussion section, we addressed the point that “we observed that the surrogate marker of gut leakage, namely iFABP concentrations, were actually lower in meningitis patients than in healthy volunteers. This finding has also been reported in the context of COVID-19 and pneumococcal pneumonia (3,4)” Other studies have shown higher concentrations of iFABP in healthy volunteers compared to patients with mesenteric ischemia, a condition associated with intestinal injury (5). While one might hypothesize an analytical error, the concentrations obtained in our healthy volunteers (432 (248-651) pg/ml) fall within the known average ranges (i.e., 200–600 pg/ml). The question of differences between the assay tools used can also be raised. Nevertheless, some authors have validated these observations using multiple techniques (5). iFABP levels in some healthy individuals may reflect non-pathological individual variations, such as oxidative stress or age-related factors. However, age was balanced between the two groups, and no underlying diseases causing intestinal ischemia were identified in healthy individuals. Finally, the timing of measurements could explain some discrepancies in results between studies. iFABP is an early marker of intestinal epithelial damage; however, its concentration may decrease as the damage resolves, which can result in variability depending on the timing of sample collection. In our study, samples from meningitis patients were collected as soon as the pneumococcal strain was identified, approximately two days after admission. We cannot rule out the possibility of an earlier increase in iFABP levels occurring immediately upon admission, followed by a subsequent decrease.

“In addition, we observed that the surrogate marker of gut leakage, namely iFABP concentrations, were actually lower in meningitis patients than in healthy volunteers. This has also been reported in the setting of COVID-19 and pneumococcal pneumonia (9,13). Other studies have also demonstrated higher concentrations of iFABP in healthy volunteers compared to patients with mesenteric ischemia, a condition well-known for its association with intestinal injury (14). Additionally, no underlying diseases causing intestinal ischemia were identified in healthy individuals. However, these discrepancies between studies could be attributed to the different tools used to measure iFABP, or the timing of the iFABP measurement. In our study, sample from meningitis patients were collected as soon as the pneumococcal strain was identified, approximately two days after admission. We cannot rule out the possibility of an earlier increase in iFABP levels occurring immediately upon admission, followed by a subsequent decrease, as iFABP concentrations may decline with the resolution of intestinal damage.”

3. "Are you sure that there is no disease that causes intestinal ischemia in individuals defined as healthy individuals? If you are sure, please state this in the material method."

o Response: We updated the Methods section to specify : “Comorbid conditions, including those that could lead to intestinal ischemia, were rigorously assessed.” And in the results section : “and 34 HV, none of whom had conditions predisposing to intestinal ischemia.”

Response to Reviewer #2:

1. General comment – This is a very interesting research question that aims to define a role, if any, of LPS in the systemic inflammatory response during pneumococcal meningitis. The study is well written and diligently conducted. The results are significant and relevant, adding good information for understanding the pathophysiology of this disease. There are some points the authors should review, as follows

o Response: We thank the Reviewer 2 for these positive and kind comments.

2. "In Introduction section correct, in Line 3, Neisseria meningitidis (N. meningitidis)"

o Response: The requested correction has been made in the Introduction section.

3. "Which was the time frame for including healthy volunteers? Describe it in the manuscript."

o Response: We added to the Methods section that healthy volunteers were included between April and June 2022.

4. "The dried extract extracts - remove the duplicate word"

o Response: The typographical error has been corrected.

5. " Then, we observed a difference between patients and volunteers, considering immunocompromised status, which may introduce a potential bias in the interpretation of plasma endotoxin concentrations. Since immunocompromised patients are susceptible to gut transloca

---

## [Decision Letter · Decision Letter 1]

22 Apr 2025

Pneumococcal meningitis and endotoxemia: a cross-sectional clinical study

PONE-D-24-44702R1

Dear Dr. Blot,

We’re pleased to inform you that your manuscript has been judged scientifically suitable for publication and will be formally accepted for publication once it meets all outstanding technical requirements.

Kind regards,

Muhammad Afzal, PhD

Academic Editor

PLOS ONE

Additional Editor Comments (optional):

Reviewers' comments:

Reviewer's Responses to Questions

**Comments to the Author**

1. If the authors have adequately addressed your comments raised in a previous round of review and you feel that this manuscript is now acceptable for publication, you may indicate that here to bypass the “Comments to the Author” section, enter your conflict of interest statement in the “Confidential to Editor” section, and submit your "Accept" recommendation.

Reviewer #1: All comments have been addressed

Reviewer #2: All comments have been addressed

2. Is the manuscript technically sound, and do the data support the conclusions?

Reviewer #1: Yes

Reviewer #2: Yes

3. Has the statistical analysis been performed appropriately and rigorously? 

Reviewer #1: I Don't Know

Reviewer #2: Yes

4. Have the authors made all data underlying the findings in their manuscript fully available?

Reviewer #1: Yes

Reviewer #2: Yes

5. Is the manuscript presented in an intelligible fashion and written in standard English?

Reviewer #1: Yes

Reviewer #2: Yes

6. Review Comments to the Author

Reviewer #1: Dear authors, thank you very much, the corrections made are suitible and enough for me. Best regards

Reviewer #2: Dear authors, thank you for addressing carefully all suggestions. I do not have additional comments.

7. PLOS authors have the option to publish the peer review history of their article (what does this mean? ). If published, this will include your full peer review and any attached files.

**Do you want your identity to be public for this peer review?** For information about this choice, including consent withdrawal, please see our Privacy Policy .

Reviewer #1: No

Reviewer #2: No

---

## [Editor Report · Acceptance letter]

PONE-D-24-44702R1

PLOS ONE

Dear Dr. Blot,

I'm pleased to inform you that your manuscript has been deemed suitable for publication in PLOS ONE. Congratulations! Your manuscript is now being handed over to our production team.

Kind regards,

on behalf of

Dr. Muhammad Afzal

Academic Editor

PLOS ONE